# Can internal medicine specialists diagnose functional somatic disorders (FSDs)? Training and comparison with FSD specialists

**Michael Moesmann Madsen**[1,2,3,4]*, **Christian Trolle**[1,4], **Lotte Fynne**[1,4], **Eva Ørnbøl**[3,4], **Per Fink**[3,4], **Lise Kirstine Gormsen**[3,4]

**1** University Clinic for Innovative Patient Pathways, Diagnostic Center, Silkeborg Regional Hospital, Silkeborg, Denmark, **2** Silkeborg Psychiatric Center, Central Regional Psychiatry, Silkeborg, Denmark, **3** The Research Clinic for Functional Disorders, Aarhus University Hospital, Aarhus, Denmark, **4** Faculty of Health Sciences, Aarhus University, Aarhus, Denmark

* michms@rm.dk

## Abstract

**Data Availability Statement:** All relevant data are within the paper and its Supporting Information file.

### Background

Functional somatic disorders (FSD) are a common problem across medical settings and remain challenging to diagnose and treat. Many patients with FSD undergo sequential and unnecessary extensive diagnostic work-up, which is costly for society and stressful for patients. Previous studies have shown that the empirically based FSD diagnostic entities are interrater reliable and stable over time.

### Objective

The aim of this study was to investigate whether internists who have received adequate training and with sufficient time per patient could diagnose FSD.

### Design

This was a prospective diagnostic accuracy study. The study was conducted from May 2020 to April 2022.

### Participants

The study included 27 consecutive patients referred by their general practitioner to a non-psychiatric diagnostic clinic for assessment of physical symptoms on suspicion of FSD.

### Interventions

The internists received a 30-hour training course in the use of a tailored version of the SCAN interview.

**Funding:** MMM - TRYG Foundation (TrygFonden) Grant ID: 147304. www.tryghed.dk The TRYG Foundation had no influence on the study design, data collection, analysis, decision to publish or preparation of this manuscript.

**Competing interests:** The authors have declared that no competing interests exist.

## Main measures

The main outcome measure was the agreement between the diagnoses of the internists and the reference diagnoses made by specialists in FSD on the basis of the full SCAN interview.

## Key results

The interrater agreement between the internists and the FSD experts was substantial for any FSD (kappa = 0.63) as well as multi-organ vs. single-organ FSD (kappa = 0.73), indicating good diagnostic agreement.

## Conclusions

Internists with proper training and sufficient time (3–4 hours) per patient can proficiently diagnose FSD employing a tailored version of the SCAN interview for use in a non-psychiatric diagnostic setting.

## Introduction

Functional Somatic Disorders (FSDs) are common across medical settings [1], and recent studies have found a high prevalence of FSD (8–10%) in the general population [2, 3]. Patients with FSD predominantly present in non-psychiatric settings with multiple physical symptoms that may mimic various other physical diseases and therefore present complex differential diagnostic dilemmas. Furthermore, patients with FSD may also have physical and mental comorbidities, which increases the complexity of the diagnostic evaluation. Many patients with undiagnosed FSD are repeatedly referred for diagnostic evaluation by various specialists, leading to overutilization of diagnostic and treatment resources [4–6]. Thus, some of these patients may be on a seemingly endless "odyssey" of repeated referrals for diagnostic evaluation in subspecialty clinics for years, potentially leading to more chronic disease, psychological distress, lower labor market participation, delay in treatment, and risk of iatrogenic harm from excessive diagnostic procedures/interventions [7].

A wide range of terms have been employed to denote these disorders, such as medically unexplained symptoms, somatoform disorders, and functional somatic syndromes (FSS), including fibromyalgia, irritable bowel syndrome (IBS), chronic fatigue syndrome (CFS)/myalgic encephalopathy (ME), and multiple chemical sensitivity (MCS) [8, 9]. Based on empirical research, and equivalent to the bodily distress syndrome (BDS) research diagnosis, a functional somatic disorder phenotype has been identified [9]. FSDs are characterized by identifiable, persistent and bothersome physical symptom patterns from one (single-organ FSD) or several (multi-organ FSD) of four organ groups: cardiopulmonary (CP), gastrointestinal (GI), musculoskeletal (MS), and general symptoms (GS) group [8, 10]. As with all other clinical diagnoses, relevant differential diagnoses must have been considered.

General practitioners find that this patient group is among the most difficult to manage [11–14]. Many patients with undiagnosed FSD are repeatedly referred for specialized diagnostic evaluation, leading to consternation among some clinicians [4, 15, 16]. From a societal perspective, the FSD patient group is overly costly due to excess use of medical care, lower labor market participation, and lost working years [17].

In Denmark, the Danish Health Authority (Sundhedsstyrelsen) has mandated that FSD cases of mild and moderate severity should be managed in primary care, and only severe FSD cases should be managed at specialized FSD clinics [18]. This implies that most FSD patients have to be diagnosed and treated in primary care with the support of FSD specialists.

Since 2008, all trainee General Practitioners (GPs) in Western Denmark have received basic training in diagnosing FSD and communicating/negotiating the diagnosis with FSD patients using The Extended Reattribution and Management (TERM) model [16, 19]. However, many GPs feel inadequate in managing the more complicated cases, especially at the early stage where the diagnosis is still uncertain and physical differential diagnoses have not been excluded. Furthermore, the current general medicine diagnostic centers still have inadequate knowledge of FSD and are thus unable to sufficiently assist GPs in obtaining diagnostic certainty for patients with mild or moderate FSD.

To address this gap in diagnostic availability, it was decided to set up a Diagnostic Clinic for Functional Disorders (FSD clinic) at the general internal medicine Diagnostic Center (DC) at Silkeborg Regional Hospital. DC Silkeborg encompasses all internal medicine subspecialties, including radiology. As in other diagnostic centers, DC Silkeborg offers a patient-centric, multidisciplinary "same-day diagnosis" approach, where patients receive a comprehensive diagnostic evaluation and results within the same day in order to swiftly identify and address serious medical conditions [20, 21]. The new FSD Clinic at DC Silkeborg is being evaluated in a randomized clinical trial called the DISTRESS Trial [22]. As a novelty at this clinic, internal medicine specialists, after being trained in FSD assessment and patient education for FSD by FSD experts, performed the FSD diagnostic work using a tailored version of the SCAN interview [16, 19]. This approach transfers knowledge from a specialized FSD department, traditionally staffed mainly by psychiatrists, to a physical diagnostic center setting staffed by internal medicine specialists. In this way, patients presenting with physical symptoms at a physical diagnostic center are assessed using a customized SCAN interview similar to those seen at the highly specialized FSD center. To our knowledge, this has never been attempted before, and thus there is a clear gap in our knowledge regarding the feasibility of internal medicine specialists (internists) to diagnose FSD in a clinical setting.

In the present study, we aimed to evaluate the feasibility of training internists in carrying out clinical assessments of FSD as well as compare their diagnostic outcomes with those from gold standard interviews by experienced FSD clinicians (FSD specialists).

## Methods

### Study population and study design

Participants in the DISTRESS Trial were recruited from the Central Denmark Region, comprising ~1.2 million people. To be included in the DISTRESS Trial, patients had to be referred by their GP for diagnostic evaluation due to symptoms presenting a diagnostic dilemma between well-defined physical diseases and a suspected FSD. Additionally, for inclusion, patients had to be between 18 and 60 years old, speak and read Danish fluently, and to have had suspected symptoms of FSD for at least 6 months and no more than 3 years. Patients were excluded if they had a pre-existing severe chronic physical disease that explained their reduction in level of functioning in daily life, or if they had previously been evaluated at a specialized FSD clinic. Patients were also excluded if they were pregnant, were abusing alcohol or non-prescription drugs, or had an acute or severe psychiatric disorder, such as psychosis, bipolar affective disorder, or severe depression with psychotic symptoms.

After inclusion in the DISTRESS Trial, patients were randomized 1:1 to either the intervention group, and subsequently seen at the FSD Clinic at DC Silkeborg by our internists trained

in FSD diagnostics and patient education, or to diagnostic as usual, where they were offered an alternative diagnostic evaluation at another pre-existing specialist clinic. The choice of the most relevant alternative diagnostic evaluation was determined by their GP.

For the present study, 27 consecutive patients from the intervention group of the DISTRESS Trial were invited to a subsequent SCAN Gold Standard interview by an experienced FSD clinician from a specialized FSD clinic with substantial experience in diagnosing FSD including the use of the SCAN interview. The present study started recruitment on 13 May 2020 and was completed on 21 April 2022.

## Training the internists

Seven internists received training from FSD specialists and then carried out clinical patient visits including a tailored SCAN and an FSD-focused interview at the new FSD Clinic at DC Silkeborg. The seven internists were from the internal medicine subspecialties: rheumatology (N = 4), endocrinology (N = 2), and gastrointestinal medicine (N = 1). None of the trainees had had previous training in FSD diagnostics and patient education. Training was carried out by three FSD specialists (psychiatrists) at the specialized department for Functional Disorders at Aarhus University Hospital. The training consisted of a four-day, 26 hour in-person initial residential training course in March 2019, followed by four biannual follow-up 1-day seminars that took place from October 2019 to November 2021 with regular supervision by an FSD specialist. The training schedule was somewhat extended due to the COVID-19 pandemic. Furthermore, the internists attended one SCAN interview performed by an FSD specialist. Additionally, bimonthly Q&A sessions with the principal investigator of this project were held. The training was a multifaceted educational program (Table 1). The course content included: 1) The assessment, treatment, and management of FSD, i.e., the TERM model with hands-on role play including micro-skill training using professional actors, and 2) An introduction to the principles of SCAN Rating and hands-on training in a tailored version of SCAN. The internists were trained in the entire FSD diagnostic process, including biopsychosocial history-taking, the SCAN interview to identify the positive diagnostic criteria for FSD and to identify psychiatric or general medicine comorbid/differential diagnostic conditions, as well as in the FSD communication approach using TERM [19, 23]. The trainees received subsequent training and calibration through consensus conversations with FSD specialists as described below in the "SCAN re-interview procedure" section (Table 1).

## Instruments

Prior to randomization in the DISTRESS trial, participants filled out a questionnaire, including a battery of patient-reported outcomes (Fig 1). Specifically, the battery included the BDS Checklist [24], Whiteley-6R [25] health anxiety index, and several symptom checklists: SCL-8 [26], SCL-4anx [27], and SCL-6dep [28].

A tailored version of the semi-structured online-based computer-assisted SCAN 3.0 for use by internists for FSD diagnostics in a physical diagnostic setting was used. The SCAN version 2.1 is among the most widely used and broadly validated instruments for diagnosis in neuropsychiatry. The SCAN 3.0 is a new and updated version of SCAN and includes, i.a., a second section on functional disorders, physical disease, and health anxiety as well as on conversion and dissociative disorders that has been developed to replace the old section on somatoform and related conditions. The online-based electronic version was recently developed. The tailored version is a reduced version of the full SCAN 3.0, e.g., within affective disorders it only includes the key symptoms of depression and within anxiety only the screening questions, and psychotic disorders are not included. After symptom rating by the clinician rater, the SCAN

**Table 1. Internist training program in FSD diagnostics and patient education for FSD.**

| |
|---|
| **Initial training:** |
| *Residential course (3 days x 6.5 hours)* |
| • Didactic sessions: theory and empirical background of functional somatic disorders as diagnostic entity |
| • Workshop on the participants' experiences with Functional somatic disorders (FSD) |
| FSD diagnostic training |
| • Hands-on training in our tailored electronic version of SCAN 3.0 |
| • Six sessions of internal medical specialists' diagnostic consultations performed with professional actors, videotaped, and subsequently supervised in groups of four with FSD specialists. |
| Patient education for FSD |
| • Stepwise presentation of TERM and introduction to exercises |
| • Six videotaped modules of micro-skills training in pairs as well as with professional actors. |
| • Further dialogue focused on incentives, barriers, strengths, weaknesses, opportunities, and threats concerning TERM applied in clinical practice. |
| • Two rounds of group discussions |
| *Follow-up session (6.5 hrs–two weeks after the residential course)* |
| • Didactic sessions: theory and empirical basis of SCAN |
| • Expert video of FSD patient consultation |
| • Further training in the tailored electronic version of SCAN 3.0. |
| • Video supervision of internists' patient consultations including FSD diagnostic interviews and patient education for FSD. |
| • Interrater SCAN rating session with FSD specialists. Patient interview supervised by FSD specialists. |
| **Total time used on the initial training program by each participating physician including reading and video preparation: 30 hrs** |
| *Follow-up training* |
| • Attending one SCAN interview performed by an FSD specialist. |
| • Bi-annual one-day seminars including SCAN interrater sessions. |

3.0 software algorithm identifies any FSD as well as FSD single- vs. multi-organ if present. The SCAN 3.0 software also identifies somatoform disorders as well as specialty-specific syndrome diagnoses (FSS) such as fibromyalgia, irritable bowel disease, and CFS/ME using various criteria including the CDC and Oxford criteria.

## SCAN re-interview procedure

The reinterview, based on the gold standard SCAN interview, was conducted by an experienced FSD clinician (FSD specialist) from a highly specialized FSD center as soon as possible after the first interview. The FSD specialist (LG) used the full SCAN interview, excluding the chapters on psychosis. The FSD specialist was blinded to the evaluation of the internist interviews, and she received all relevant clinical data up to the time when the internist had seen the patient.

The reinterview consultation lasted three hours and had almost the same format as the initial diagnostic visit with the internist: one hour for preparation including review of patient records, information from the GP, and the baseline questionnaire followed by the patient visit, in which one hour was dedicated to biopsychosocial history taking and one hour to the SCAN interview (Fig 1). As at the initial diagnostic visit, a diagnosis of FSD was made where applicable and documented based on the SCAN interview, informed by the SCAN software algorithm's output.

Subsequently, a consensus discussion was held between the "primary" SCAN interviewer and the FSD specialist. After the FSD specialist had documented and revealed her final SCAN interview conclusion, the evaluation including the conclusion from the internist SCAN

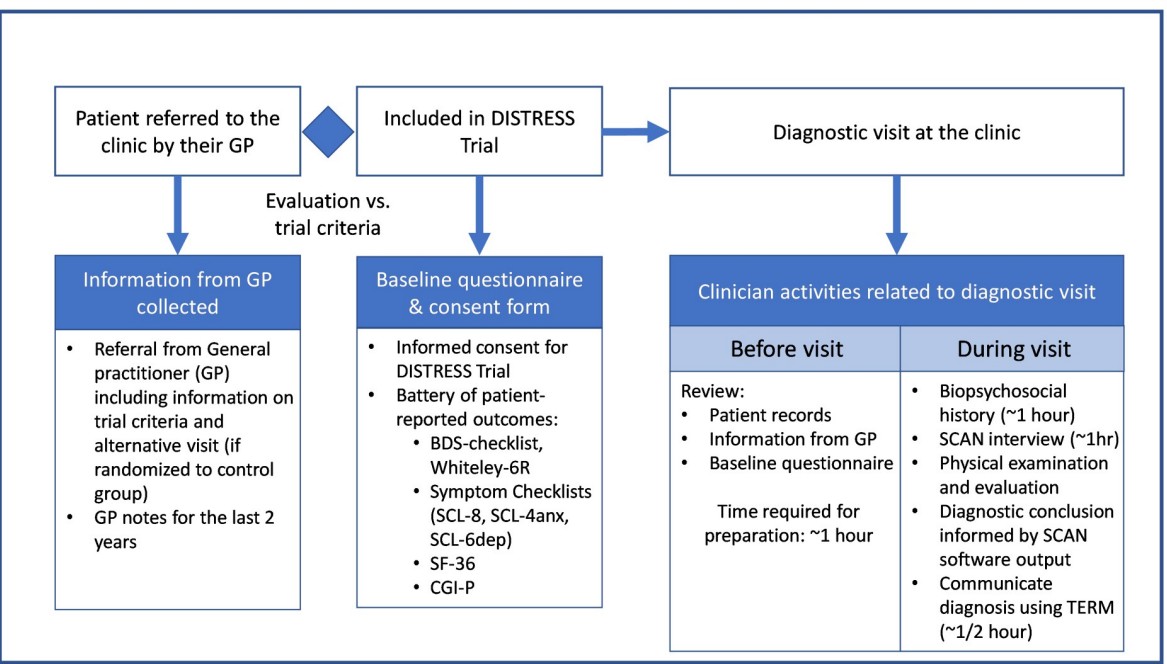

**Fig 1. Diagnostic clinic flow.**

interview was unblinded, and a consensus conversation was held between the FSD specialist and the internist.

## Statistical analysis

The FSD specialist's SCAN interview outcome was regarded as the gold standard, and comparisons were made examining the extent of agreement between the FSD specialist and the internists. A two-level hierarchy of diagnostic boundaries was applied for the evaluation of the accuracy of the diagnostic outcomes of the internists' interviews. The first level was a two-fold classification: absence or presence of FSD, and the second level was single-organ vs. multi-organ FSD. At each level, 2x2 tables were constructed. We measured the agreement between the internists and the FSD specialist by inter-observer agreement as well as by Cohen's Kappa [29]. Landis and Koch's division into classes of agreement was used for interpretation of inter-rater reliability [30]: 0.00–0.20 slight agreement, 0.21–0.40 fair agreement, 0.41–0.60 moderate agreement, 0.61–0.80 substantial agreement, and 0.81–1.00 near perfect agreement. We also calculated the sensitivity, specificity, positive predictive value (PPV), and negative predictive value (NPV) of the internist ratings versus the assessments by the FSD specialists. For all statistical measures calculated, 95% confidence intervals were constructed. All analyses were performed in Stata 17.0 for Windows (StataCorp LLC, College Station, USA).

## Trial registration and ethical approval

The DISTRESS Trial has been registered at clinicaltrials.gov with ID NCT06025617 [22]. The Scientific Ethics Committees for the Central Denmark Region concluded in November 2018 that the project did not require ethical approval from the committee. Regarding data protection, we have sought and received approval (case number 1-16-02-160-19) from the Data

Protection Authority in the Central Denmark Region. Written informed consent for inclusion in the study was obtained from each patient prior to inclusion.

## Results

### Inclusion

In total, 27 consecutive patients from the intervention group of the DISTRESS Trial were invited to participate in Gold standard interviews. The median age of the 27 participants was 34 years (range 18–64) and 23 of the participants (85%) were female. All 27 agreed to participate, and hence all 27 participants completed interviews by both an internist and the FSD specialist.

### Evaluation of the SCAN interviews by the internists as a diagnostic test

The results of the comparison of the 27 gold standard interviews with the primary SCAN interviews by the internists are shown in Table 2. The FSD specialist found that 24 out of 27 patients assessed had any FSD (positive rate 0.89), and 16 out of 23 patients with any FSD as rated by both the Internist and the FSD specialist had multi-organ FSD (positive rate 0.70).

At the 1st diagnostic level of classification of any FSD vs. no FSD, the internists were very good at "detecting" FSD and quite good at "ruling out" FSD, having missed only one positive and one negative of the 27 patient cases compared with the gold standard interviews by the FSD specialist. At the 2nd diagnostic level of classification of single- vs. multi-organ FSD, the internists rated all the single-organ FSD cases as single-organ and 3 of 16 multi-organ FSD cases as single-organ FSD. Thus, the internists misclassified three out of 16 cases of multi-organ FSD as single-organ FSD compared to the gold standard interview.

Table 3 shows the interrater agreement, which we found to be substantial, with Kappa values of 0.63 (0.15–1.00) and 0.73 (0.45–1.00) for any FSD vs. no FSD, and single-organ vs. multi-organ FSD, respectively (Table 3). However, the confidence intervals were quite wide (especially for the 1st diagnostic level).

## Discussion

This feasibility study suggests that internists can indeed reliably diagnose FSD in a diagnostic center setting. A tailored SCAN interview was used to support the clinical diagnostic process and taught in a 30-hour intensive training course followed by supervision by a specialist in FSD. In this study, we found that internists in a non-psychiatric diagnostic center setting after

**Table 2. Diagnostic outcomes of trained internist and FSD specialist interviews.**

| | Diagnostic interview outcomes | | | | FSD-positive rate in study sample |
|---|---|---|---|---|---|
| | Both interviews negative (n) | Only internist positive (n) | Only FSD expert positive (n) | Both interviews positive (n) | FSD specialist |
| | -/- | -/+ | +/- | +/+ | # |
| Any FSD | 2 | 1 | 1 | 23 | 24/27 |
| Multi-organ FSD (vs. single-organ) | 7 | 0 | 3 | 13 | 16/23* |

Abbreviations: FSD = Functional Somatic Disorder.

*Positive rate of Multi-organ FSD among patients where both interviewers found any FSD.

**Table 3. Agreement between SCAN-trained internists and FSD specialist diagnoses.**

|  | Cohens Kappa (95% CI) | Observed agreement % (95% CI) | Sensitivity (95% CI) | Specificity (95% CI) | PPV (95% CI) | NPV (95% CI) |
|---|---|---|---|---|---|---|
| Any FSD | 0.63 (0.15–1.00) | 92.5 (75.7–99.1) | 95.8 (78.9–99.9) | 66.7 (9.4–99.2) | 95.8 (78.9–99.9) | 66.7 (9.4–99.2) |
| Multi-organ FSD (vs. single-organ) | 0.73 (0.45–1.00) | 87.5 (67.6–97.3) | 81.3 (54.4–96.0) | 100 (59.0–100.0) | 100 (75.3–100.0) | 70 (34.8–93.3) |

Abbreviations: CI = Confidence interval (analytic); PPV = positive predictive value; NPV = Negative predictive value

receiving training in FSD diagnostics and patient education for FSD were good at identifying FSD, with substantial agreement compared with FSD specialists.

To our knowledge, this is the first time internists have been systematically trained in how to identify FSD using the SCAN interview as a diagnostic aid for clinical FSD diagnostics. The substantial interrater agreement is perhaps further remarkable given that we not only transferred the ability to conduct SCAN from one group of specialists to another, but also from a highly specialized clinical setting to a general internal medicine diagnostic center in the secondary sector.

There was a high positive rate of FSD among our interview subjects, which was expected, since patients were recruited from the DISTRESS trial–in which subjects are included after being referred by their GP to our clinic based on suspicion of FSD by their GP. Due to the low number of interviews and the high positive rate of FSD in our study sample, we obtained wide confidence intervals on our measurements of agreement, however, even considering this uncertainty, the interrater agreement between the trained internists and the FSD specialist was substantial.

The extent of agreement achieved at both levels of the FSD diagnostic hierarchy is on par with those found in the WHO PSE-10 SCAN field trials [31]. When considering any generalization to other diagnostic centers, we should highlight that the internists in this study had substantial time allocated to each clinical visit for the interviews (3–4 hours per patient) equivalent to the time dedicated at the highly specialized FSD department in Aarhus.

Not one case of single-organ FSD was misclassified as multi-organ FSD, however in three of 16 instances where the internist concluded single-organ FSD, the gold standard interviewer concluded multi-organ FSD. These misclassifications occurred throughout the study period and thus cannot be ascribed to the learning curve. The misclassifications were not associated with individual internist subspecialties, since symptoms from the cardiac symptom cluster were overlooked in two of the cases, whereas, general symptoms were overlooked in the final case. A possible explanation could be that the internists were more prone to judge specific symptoms as pertaining to other causes than FSD, or that they were more likely to overlook some FSD symptoms. Furthermore, although the FSD specialist and the internists rated the same period for the SCAN interviews, patients' responses may have differed between the interviews due to repeated questioning, increased symptom awareness, or recall bias due to the time passed between the two interviews.

In any test/retest study, the amount of time elapsed between two interviews can reduce concordance between the test and retest interview. The median time between the two interviews in our study was 42 days [23–52 IQR] due to scheduling constraints by the FSD specialist as well as interruptions due to COVID restrictions during the study period. However, this time delay arguably lends more credence to our finding of substantial interrater agreement, since if

the time delay did introduce a bias, it would be a bias towards greater disagreement between the two interview results.

## Strengths and limitations

The strengths of this study include the prospective design where all invited patients agreed to participate and completed both interviews, increasing the validity of our findings. Furthermore, the internists' diagnoses were compared to an established gold standard of FSD specialist assessments. Also, the study was undertaken in a non-psychiatric clinical diagnostic center setting and thus arguably has relevance to clinical practice in similar settings.

A notable limitation of this study is the feasibility of dedicating 3–4 hours per patient in one day as practiced in our study. While this approach aligns with the same-day diagnostic models already established at DC Silkeborg and at other diagnostic centers, it may be difficult to implement in all clinical settings. However, the clinicians at the FSD clinic assessed that shortening the duration of each visit was not feasible and argued that the extensive time spent was entirely necessary. They noted that reducing the time per visit would require patients to attend multiple sessions, resulting in an equal or greater overall time commitment. The cost-effectiveness of this comprehensive visit model is currently being evaluated in the DISTRESS trial [22], however, adapting this model to conventional practice elsewhere may require significant adjustments in scheduling and resource allocation.

The present study is also limited by the small sample size (N = 27). Thus, to fully evaluate the validity of our approach, a larger study would be needed. Furthermore, since all patients in this study were recruited after being referred by their GP on suspicion of FSD, the pretest positive rate in the included population was high. In summary, we cannot conclude on the generalizability to other clinical settings or to patient populations not specifically referred by their GP on suspicion of having an FSD based on these data.

## Conclusions

The results suggest that with proper training and sufficient time per patient, internists can reliably diagnose FSD using a tailored version of the SCAN interview in a general physical diagnostic center setting.

## Supporting information

**S1 Dataset.**
(XLSX)

## Acknowledgments

We would like to express our deep appreciation for our project nurses, Anna Sofie Mensberg and Anne Sofie Bøtcher Glenting, as well as our data manager, Andrew Bolas and our software developer, Philip Anthony Riley. We would also like to thank our clinicians Michele Colombo, Rasmus Lederballe Pedersen, Susan Ringskær Christensen and Vibeke Neergaard Sørensen, and. Our thanks also to Helle Obenhausen Andersen for proofreading the manuscript.

## Author Contributions

**Conceptualization:** Michael Moesmann Madsen, Lise Kirstine Gormsen.

**Data curation:** Michael Moesmann Madsen, Eva Ørnbøl.

**Formal analysis:** Michael Moesmann Madsen, Eva Ørnbøl.

**Funding acquisition:** Michael Moesmann Madsen.

**Investigation:** Michael Moesmann Madsen, Christian Trolle, Lotte Fynne,
Lise Kirstine Gormsen.

**Methodology:** Michael Moesmann Madsen, Lise Kirstine Gormsen.

**Project administration:** Michael Moesmann Madsen.

**Resources:** Eva Ørnbøl, Per Fink.

**Supervision:** Christian Trolle, Per Fink, Lise Kirstine Gormsen.

**Validation:** Eva Ørnbøl.

**Visualization:** Michael Moesmann Madsen.

**Writing – original draft:** Michael Moesmann Madsen.

**Writing – review & editing:** Michael Moesmann Madsen, Christian Trolle, Lotte Fynne,
Per Fink, Lise Kirstine Gormsen.

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
