## [Decision Letter · Decision Letter 0]

26 Mar 2024

PONE-D-23-38611Can Internal Medicine specialists diagnose Functional Somatic Disorders (FSDs)? - Training and comparison with FSD specialists.PLOS ONE

Dear Dr. Madsen,

Thank you for submitting your manuscript to PLOS ONE. After careful consideration, we feel that it has merit but does not fully meet PLOS ONE’s publication criteria as it currently stands. Therefore, we invite you to submit a revised version of the manuscript that addresses the points raised during the review process.

 The contributions of this study must be highlighted in the introduction section. 

We look forward to receiving your revised manuscript.

Kind regards,

Zulqarnain Mushtaq, PhD

Academic Editor

PLOS ONE

Journal Requirements:

Reviewers' comments:

Reviewer's Responses to Questions

**Comments to the Author**

1. Is the manuscript technically sound, and do the data support the conclusions?

Reviewer #1: Yes

Reviewer #2: Yes

2. Has the statistical analysis been performed appropriately and rigorously? 

Reviewer #1: Yes

Reviewer #2: Yes

3. Have the authors made all data underlying the findings in their manuscript fully available?

Reviewer #1: Yes

Reviewer #2: Yes

4. Is the manuscript presented in an intelligible fashion and written in standard English?

Reviewer #1: Yes

Reviewer #2: Yes

5. Review Comments to the Author

Reviewer #1: The theme of the topic is good, and the paper provides a thorough literature review and background study. Still, the paper has some hindrances in justifying the study in the introduction section, the importance of selected models in the literature section, and the selected model should be validated in the methods and data section. I suggest the publication, but after considering the following minor comments.

1: Study Design: when was the study conducted?

2. The Introduction Section needs to be improved with relevant and updated references.

3: Time per patient: While the abstract mentions that internists require 3-4 hours per patient, it does not provide information on the practicality or feasibility of this time requirement in real-world clinical settings.

4: Generalizability: The study participants were referred to a non-psychiatric diagnostic clinic, which may limit the generalizability of the findings to other settings or populations.

5: Small sample size: The study included only 27 patients, which may limit the generalizability of the findings. A larger sample size would provide more robust results and increase confidence in the conclusions drawn.

6. Check the presentation and Grammar of the article.

7: Improve the quality of your figure in the manuscript.

Reviewer #2: Dear authors

This is a well-written, sound and original paper presenting a new way of diagnosing patients with severe and complex symptoms known as functional somatic symptoms. The idea of the project great and the results very promising for this new model. Congratuations!!

The methods applies are well-applied and the statistics fine and reasonable. The only concern is the amount of participants, which is also mentioned as the weak point in this study,

I have no suggestions for major revision, only one comment in "Limitations" line 297-298:

"Limitations of this study include the small sample size (N=27) (which is arguably a large sample

given the specialist time requirements)".

I would suggest to leave out the comment in bracket not trying to justify the small sample size, but keep it simple

6. PLOS authors have the option to publish the peer review history of their article (what does this mean?). If published, this will include your full peer review and any attached files.

Reviewer #1: No

Reviewer #2: No

---

## [Author Response · Author response to Decision Letter 0]

13 May 2024

Response to Reviewer Comments - PONE-D-23-38611 - "Can Internal Medicine specialists diagnose Functional Somatic Disorders (FSDs)? - Training and comparison with FSD specialists"

Dear Dr. Mushtaq,

We would like to express our sincere gratitude to the reviewers for their constructive comments and to you for the opportunity to improve our manuscript. We have carefully considered each point raised and have made corresponding revisions to the manuscript. Please find below a detailed response to each of the comments:

Study Design (Comment on dates of the study): “Study Design: when was the study conducted?”

The dates are listed in the study in the Methods section lines 122-123 (previous version of the manuscript). However, to clarify and highlight the temporal scope within which our research was conducted, we have also added “The study was conducted between May 2020 and April 2022” to the abstract.

Introduction Section (Need for updated references): “The Introduction Section needs to be improved with relevant and updated references.”

Thank you for your valuable feedback regarding the need for updated references in the Introduction section of our manuscript. In response to your comments, we have added several additional relevant references. If there are specific recent articles you believe should be included to enhance our introduction, please feel free to mention them for our consideration.

Time per Patient (Practicality and feasibility): “Time per patient: While the abstract mentions that internists require 3-4 hours per patient, it does not provide information on the practicality or feasibility of this time requirement in real-world clinical settings.”

We have added a discussion in the Limitations section (now also reflected in the Introduction) about the practical challenges of implementing extended visit times in the context of to the “same-day diagnostic” approach used in this and other diagnostic centres, where extended consultations can lead to more efficient patient management, albeit requiring adjustments in scheduling and resource allocation. Additionally, the economic impact of this approach is being assessed in the ongoing DISTRESS trial (NCT06025617), providing crucial data on its cost-effectiveness.

Generalizability (Referral to a non-psychiatric diagnostic clinic): “The study participants were referred to a non-psychiatric diagnostic clinic, which may limit the generalizability of the findings to other settings or populations.”

Thank you for your comment regarding the generalizability of our findings. We have adapted the introduction to reflect the fact that patients included in our study are typically managed within general (physical) medicine diagnostic centres. Given that these patients present with physical symptoms and physical diagnostic uncertainty remains, these patients are rarely if ever referred to psychiatric departments. By transferring knowledge from the highly specialized clinic for functional disorders to a general physical (general medicine) diagnostic center, we thus arguably augment the provision of the right level of care in the most appropriate setting, enhancing the relevance and applicability of our findings to similar healthcare systems.

Small Sample Size (Limitations on generalizability): “Small sample size: The study included only 27 patients, which may limit the generalizability of the findings. A larger sample size would provide more robust results and increase confidence in the conclusions drawn.”

Acknowledging the limitation posed by our sample size, we have refined our discussion in the Limitations section. We removed the qualifying parenthesis and emphasized the need for further research with larger sample sizes to enhance the robustness of the findings.

Presentation and Grammar: “Check the presentation and Grammar of the article.”

We have submitted the manuscript for a thorough language review to ensure the clarity and grammatic correctness of the text.

Quality of Figure: “Improve the quality of your figure in the manuscript.”

We have used the PACE preflight system to improve the figure's quality. Do let us know if we need to revise the figure additionally.

Attached, please find the revised manuscript (marked-up and clean versions) and the updated figure as per PLOS ONE's guidelines. We believe that these revisions have substantially improved our manuscript and hope that it is now suitable for publication in PLOS ONE.

Thank you for considering our revised manuscript. We appreciate the chance to make these improvements and look forward to your decision.

Sincerely,

Michael Moesmann Madsen

---

## [Decision Letter · Decision Letter 1]

28 Jun 2024

Can Internal Medicine specialists diagnose functional somatic disorders (FSDs)? Training and comparison with FSD specialists

PONE-D-23-38611R1

Dear Dr. Madsen,

We’re pleased to inform you that your manuscript has been judged scientifically suitable for publication and will be formally accepted for publication once it meets all outstanding technical requirements.

Kind regards,

Zulqarnain Mushtaq, PhD

Academic Editor

PLOS ONE

Additional Editor Comments (optional):

Reviewers' comments:

Reviewer's Responses to Questions

**Comments to the Author**

1. If the authors have adequately addressed your comments raised in a previous round of review and you feel that this manuscript is now acceptable for publication, you may indicate that here to bypass the “Comments to the Author” section, enter your conflict of interest statement in the “Confidential to Editor” section, and submit your "Accept" recommendation.

Reviewer #1: All comments have been addressed

2. Is the manuscript technically sound, and do the data support the conclusions?

Reviewer #1: Yes

3. Has the statistical analysis been performed appropriately and rigorously? 

Reviewer #1: Yes

4. Have the authors made all data underlying the findings in their manuscript fully available?

Reviewer #1: Yes

5. Is the manuscript presented in an intelligible fashion and written in standard English?

Reviewer #1: Yes

6. Review Comments to the Author

Reviewer #1: The Authors incorporated all the suggested changes in a well-organized manner therefore, I recommend accepting the current manuscript in this form.

7. PLOS authors have the option to publish the peer review history of their article (what does this mean?). If published, this will include your full peer review and any attached files.

Reviewer #1: No

---

## [Editor Report · Acceptance letter]

4 Jul 2024

PONE-D-23-38611R1 

PLOS ONE

Dear Dr. Madsen, 

I'm pleased to inform you that your manuscript has been deemed suitable for publication in PLOS ONE. Congratulations! Your manuscript is now being handed over to our production team.

Kind regards, 

on behalf of

Dr. Zulqarnain Mushtaq 

Academic Editor

PLOS ONE